# Shallow RNNs: A Method for Accurate Time-series Classification on Tiny Devices

**Don Kurian Dennis***
Carnegie Mellon University

**Durmus Alp Emre Acar**
Boston University

**Vikram Mandikal**[†]
University of Texas at Austin

**Vinu Sankar Sadasivan**[†]
IIT Gandhinagar

**Harsha Vardhan Simhadri**
Microsoft Research India

**Venkatesh Saligrama**
Boston University

**Prateek Jain**
Microsoft Research India

## Abstract

Recurrent Neural Networks (RNNs) capture long dependencies and context, and hence are the key component of typical sequential data based tasks. However, the sequential nature of RNNs dictates a large inference cost for long sequences even if the hardware supports parallelization. To induce long-term dependencies, and yet admit parallelization, we introduce novel shallow RNNs. In this architecture, the first layer splits the input sequence and runs several independent RNNs. The second layer consumes the output of the first layer using a second RNN thus capturing long dependencies. We provide theoretical justification for our architecture under weak assumptions that we verify on real-world benchmarks. Furthermore, we show that for time-series classification, our technique leads to substantially improved inference time over standard RNNs without compromising accuracy. For example, we can deploy audio-keyword classification on tiny Cortex M4 devices (100MHz processor, 256KB RAM, no DSP available) which was not possible using standard RNN models. Similarly, using ShaRNN in the popular Listen-Attend-Spell (LAS) architecture for phoneme classification [4], we can reduce the lag in phoneme classification by 10-12x while maintaining state-of-the-art accuracy.

## 1 Introduction

We focus on the challenging task of time-series classification on tiny devices, a problem arising in several industrial and consumer applications [25, 22, 30], where tiny edge-devices perform sensing, monitoring and prediction in a limited time and resource budget. A prototypical example is an interactive cane for people with visual impairment, capable of recognizing gestures that are observed as time-traces on a sensor embedded onto the cane [24].

Time series or sequential data naturally exhibit temporal dependencies. Sequential models such as RNNs are particularly well-suited in this context because they can account for temporal dependencies by attempting to derive relations from the previous inputs. Nevertheless, directly leveraging RNNs for prediction in constrained scenarios mentioned above is challenging. As observed by several authors [28, 14, 29, 9], the sequential nature by which RNNs process data fundamentally limits parallelization leading to large training and inference costs. In particular, in time-series classification, at inference time, the processing time scales with the size, $T$, of the receptive window, which is unacceptable in resource constrained settings.

A solution proposed in literature [28, 14, 29, 9] is to replace sequential processing with parallelizable feed-forward and convolutional networks. A key insight exploited here is that most applications require relatively small receptive window, and that this size can be increased with tree-structured networks and dilated convolutions. Nevertheless, feedforward/convolutional networks utilize substantial working memory, which makes them difficult to deploy on tiny devices. For this reason, other methods such as [32, 2] also are not applicable for our setting. For example, a standard audio keyword detection task with a relatively modest setup of 32 conv filters would itself need a working memory of 500KB and about 32X more computation than a baseline RNN model (see Section 5).

**Shallow RNNs.** To address these challenges, we design a novel layered RNN architecture that is parallelizable/limited-recurrence while still maintaining the receptive field length ($T$) and the size of the baseline RNN. Concretely, we propose a simple 2-layer architecture that we refer to as ShaRNN. Both the layers of ShaRNN are composed of a collection of shallow recurrent neural networks that operate independently. More precisely, each sequential data point (receptive window) is divided into independent parts called *bricks* of size $k$, and a *shared RNN* operates on each brick independently, thus ensuring a small model size and short recurrence. That is, ShaRNN's bottom layer restarts from an initial state after every $k << T$ steps, and hence only has a short recurrence. The outputs of $T/k$ parallel RNNs are input as a sequence into a second layer RNN, which then outputs a prediction after $T/k$ time. In this way, for $k \approx O(\sqrt{T})$ we obtain a speedup of $O(\sqrt{T})$ in inference time in the following two settings:

(a) **Parallelization:** here we parallelize inference over $T/k$ independent RNNs thus admitting speed-ups on multi-threaded architectures,

(b) **Streaming:** here we utilize receptive (sliding) windows and reuse computation from older sliding window/receptive fields.

We also note that, in contrast to the proposed feed-forward methods or truncated RNN methods [23], our proposal admits fully receptive fields and thus does not result in loss of information. We further enhance ShaRNN by combining it with the recent MI-RNN method [10] to reduce the receptive window sizes; we call the resulting method *MI-ShaRNN*.

While a feedforward layer could be used in lieu of our RNN in the next layer, such layers lead to significant increase in model size and working RAM to be admissible in tiny devices.

**Performance and Deployability.** We compare the two-layer MI-ShaRNN approach against other state-of-art methods, on a variety of benchmark datasets, tabulating both accuracy and budgets. We show that the proposed 2-layer MI-ShaRNN exhibits significant improvement in inference time while also improving accuracy. For example, on Google-13 dataset, MI-ShaRNN achieves $1\%$ higher accuracy than baseline methods while providing 5-10x improvement in inference cost. A compelling aspect of the architecture is that it allows for reuse of most of the computation, which leads to its deployability on the tiniest of devices. In particular, we show empirically that the method can be deployed for real-time time-series classification on devices as those based on the tiny ARM Cortex M4 microprocessor[3] with just 256KB RAM, 100MHz clock-speed and no dedicated Digital Signal Processing (DSP) hardware. Finally, we demonstrate that we can replace bi-LSTM based encoder-decoder of the LAS architecture [4] by ShaRNN while maintaining close to best accuracy on publicly-available TIMIT dataset [13]. This enables us to deploy LAS architecture in streaming fashion with a lag of 1 second in phoneme prediction and $O(1)$ amortized cost per time-step; standard LAS model would incur lag of about 8 seconds as it processes the entire 8 seconds of audio before producing predictions.

**Theory.** We provide theoretical justification for the ShaRNN architecture and show that significant parallelization can be achieved if the network satisfies some relatively weak assumptions. We also point out that additional layers can be introduced in the architecture leading to hierarchical processing. While we do not experiment with this concept here, we note that, it offers potential for exponential improvement in inference time.

In summary, the following are our main contributions:

- We show that under relatively weak assumptions, recurrence in RNNs and consequently, the inference cost can be reduced significantly.
- We demonstrate this inference efficiency via a two-layer ShaRNN (and MI-ShaRNN) architecture that uses only shallow RNNs with a small amount of recurrence.
- We benchmark MI-ShaRNN (enhancement of ShaRNN with MI-RNN) on several datasets and observe that it learns nearly as accurate models as standard RNNs and MI-RNN. Due to limited recurrence, ShaRNN saves 5-10x computation cost over baseline methods. We deploy MI-ShaRNN model on a tiny *microcontroller* for real-time audio keyword detection, which, prior to this work, was not possible with standard RNNs due to large inference cost with receptive (sliding) windows. We also deploy ShaRNN in LAS architecture to enable streaming phoneme classification with less than 1 second of lag in prediction.

## 2    Related Work

*Stacked Architecture.* Our multi-layered RNN resembles stacked RNNs studied in the literature [15, 16, 27] but they are unrelated. The goal of Stacked RNNs is to produce complex models and subsume conventional RNNs. Each layer is fully recurrent, and feeds output of the first layer to the next level. The next level is another fully recurrent RNN. As such, stacked RNN architectures lead to increased model size and recurrence, which results in worse inference time than standard RNNs.

*Recurrent Nets (Training).* Conventional works on RNNs primarily address challenges arising during training. In particular for large receptive window $T$, RNNs suffer from vanishing and exploding gradient issues. A number of works propose to circumvent this issue in a number of ways such as Gated architectures [7, 17] or adding residual connections in RNNs [18, 1, 21] or through constraining the learnt parameters [31]. Several recent works attempt to reduce the number of gates and parameters [8, 6, 21] to reduce model size but as such suffer from poor inference time, since they are still fully recurrent. Different from these works, our focus is on reducing model size as well as inference time and view these works as complementary to our paper.

*Recurrent Nets (Inference Time).* Recent works have begun to focus on RNN inference cost. [3] proposes to learn skip connections that can avoid evaluating all the hidden states. [10] exploits domain knowledge that true signature is significantly shorter than the time-trace to trim down length of the sliding windows. Both of these approaches are complementary and we indeed leverage the second in our approach. A recent work on dilated RNNs [5] is interesting. While it could serve as a potential solution, we note that, in its original form, dilated RNN also has a fully recurrent first layer, which is therefore infeasible. One remedy is to introduce dilation in the first layer to improve inference time. But, dilation skips steps and hence can miss out on critical local context.

Finally, CNN based methods [28, 14, 29, 9, 2] allow higher parallelization in the sequential tasks but as discussed in Section 1, also lead to significantly larger working RAM requirement when compared to RNNs, thus cannot be considered for deployment on tiny devices (see Section 5).

## 3    Problem Formulation and Proposed ShaRNN Method

In this paper, we primarily focus on the time-series classification problem, although the techniques apply to more general sequence-to-sequence problems like phoneme classification problem discussed in Section 5. Let $\mathcal{Z} = \{(X_1, y_1), \ldots, (X_n, y_n)\}$ where $X_i$ is the $i$-th sequential data point with $X_i = [x_{i,1}, x_{i,2}, \ldots, x_{i,T}] \in \mathbb{R}^{d \times T}$ and $x_{i,t} \in \mathbb{R}^d$ is the $t$-th time-step data point. $y_i \in [C]$ is the label of $X_i$ where $C$ is the number of class labels. $x_{i,t:t+k}$ is the shorthand for $x_{i,t:t+k} = [x_{i,t}, \ldots, x_{i,t+k}]$.

Given training data $\mathcal{Z}$, the goal is to learn a classifier $f : \mathbb{R}^{d \times T} \to [C]$ that can be used for efficient inference, especially on tiny devices. Recurrent Neural Networks (RNN) are popularly used for modeling such sequential problems and maintain a hidden state $h_{t-1} \in \mathbb{R}^{\hat{d}}$ at the $t$-th step that is updated using:

$$h_t = R(h_{t-1}, x_t), t \in [T], \quad \hat{y} = f(h_T),$$

where $\hat{y}$ is the prediction by applying a classifier $f$ on $h_T$ and $\hat{d}$ is the dimensionality of the hidden state. Due to the sequential nature of RNN, inference cost of RNN is $\Omega(T)$ even if the hardware

supports large amount of parallelization. Furthermore, practical applications require handling a continuous stream of data, e.g., smart-speaker listening for certain audio keywords.

A standard approach is to use sliding windows (receptive field) to form a stream of test points on which inference can be applied. That is, given a stream $\mathcal{X} = [x_1, x_2, \ldots, ]$, we form sliding windows $X^s = x_{(s-1)\cdot\omega+1:(s-1)\cdot\omega+T} \in \mathbb{R}^{d\times T}$ which stride by $\omega > 0$ time-steps after each inference. RNN is then applied to each sliding window $X^s$ which implies amortized cost for processing each time-step data point $(x_t)$ is $\Theta(\frac{T}{\omega})$. To ensure high-resolution in prediction, $\omega$ is required to be a fairly small constant independent of $T$. Thus, amortized inference cost for each time-step point is $O(T)$ which is prohibitively large for tiny devices. So, we study the following key question: "*Can we process each time-step point in a data stream in $o(T)$ computational steps?*"

## 3.1 ShaRNN

Shallow RNNs (ShaRNN) are a hierarchical collection of RNNs organized at two levels. $\frac{T}{k}$ RNNs at ground-layer operate completely in parallel with fully shared parameters and activation functions, thus ensuring small model size and parallel execution. An RNN at the next level take inputs from the ground-layer and subsequently outputs a prediction.

Formally, given a sequential point $X = [x_1, \ldots, x_T]$ (e.g. sliding window in streaming data), we split it into *bricks* of size $k$, where $k$ is a parameter of the algorithm. That is, we form $T/k$ bricks: $\mathcal{B} = [B_1, \ldots, B_{T/k}]$ where $B_j = x_{((j-1)\cdot k+1):(j\cdot k)}$. Now, ShaRNN applies a standard recurrent model $\mathcal{R}^{(1)} : \mathbb{R}^{d\times k} \to \mathbb{R}^{\hat{d}_1}$ on each brick, where $\hat{d}_1$ is the dimensionality of hidden states of $\mathcal{R}^{(1)}$. That is,

$$\nu_j^{(1)} = \mathcal{R}^{(1)}(B_j), \ \ j \in [T/k].$$

Note that $\mathcal{R}^{(1)}$ can be any standard RNN model like GRU, LSTM etc. We now feed output of each layer into another RNN to produce the final state/feature vector that is then fed into a feed forward layer. That is,

$$\nu_{T/k}^{(2)} = \mathcal{R}^{(2)}([\nu_1^{(1)}, \ldots, \nu_{T/k}^{(1)}]), \quad \hat{y} = f(\nu_{T/k}^{(2)}),$$

where $\mathcal{R}^{(2)}$ is the second layer RNN and can also be any standard RNN model. $\nu_{T/k}^{(2)} \in \mathbb{R}^{\hat{d}_2}$ is the hidden-state obtained by applying $\mathcal{R}^{(2)}$ to $\nu_{1:T/k}^{(1)}$. $f$ applies the standard feed-forward network to $\nu_{T/k}^{(2)}$. See Figure 1 for a block-diagram of the architecture. That is, ShaRNN is defined by parameters $\Lambda$ composed of shared RNN parameters at the ground-level, RNN parameters at the next level, and classifier weights for making a prediction. We train the ShaRNN based on minimizing an empirical loss function over training set $\mathcal{Z}$.

Naturally, ShaRNN is an approximation of a true RNN and in principle has less modeling power (and recurrence). But as discussed in Section 4 and shown by our empirical results in Section 5, ShaRNN can still capture enough context from the entire sequence to effectively model a variety of time-series classification problems with large $T$ (typically $T \geqslant 100$). Due to parallel $k$ RNNs in the bottom layer that are processed by $\mathcal{R}^2$ in the second layer, ShaRNN inference cost can be reduced to $O(T/k + k)$ for multi-threaded architectures with $k$-wise parallelization; $k = \sqrt{T}$ leads to smallest inference cost.

**Streaming.** Recall that in the streaming setting, we form sliding windows $X^s = x_{s\cdot\omega+1:s\cdot\omega+T} \in \mathbb{R}^{d\times T}$ by striding each window by $\omega > 0$ time-steps. Hence, if $\omega = k \cdot q$ for $q \in \mathbb{N}$ then the inference cost of $X^{s+1}$ can be reduced by reusing previously computed $\nu_j^{(1)}$ vectors $\forall j \in [q+1, T/k]$ for $X^s$.

Below claim provides a formal result for the same.

**Claim 1.** *Let both layers RNNs $\mathcal{R}^{(1)}$ and $\mathcal{R}^{(2)}$ of ShaRNN have same hidden-size and per-time step computation complexity $C_1$. Then, given $T$ and $\omega$, the additional cost of applying ShaRNN to $X^{s+1}$ given $X^s$ is $O(T/k + q \cdot k) \cdot C_1$, where $X^s = x_{(s-1)\cdot\omega+1:(s-1)\cdot\omega+T}$, $\omega$ is the stride-length of sliding window, and the brick-size $\omega = q \cdot k$ for some integer $q \geqslant 1$. Consequently, the total amortized cost can be bounded by $O(\sqrt{q \cdot T}C_1)$ if $k = \sqrt{T/q}$.*

See Appendix A for a proof of the claim.

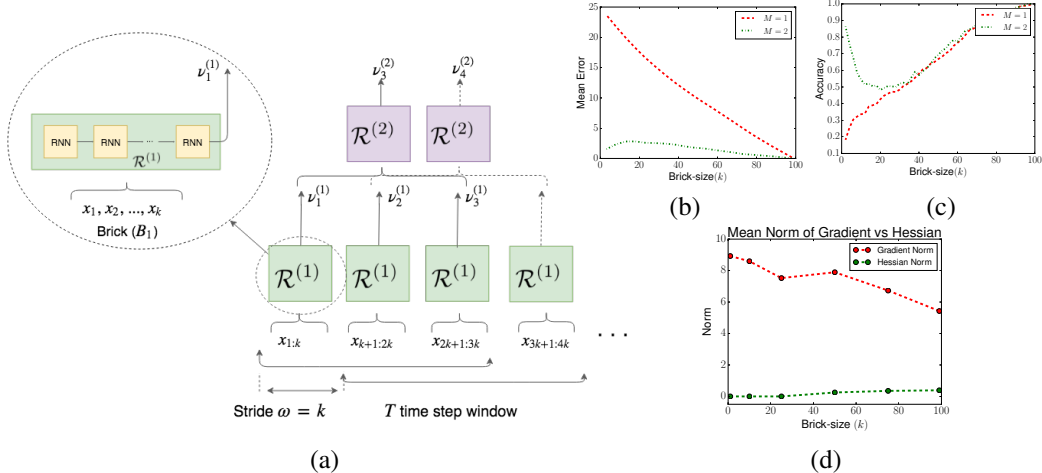

(a)                                                                    (b)                    (c)

(d)

Figure 1: **(a)** ShaRNN applies RNN $\mathcal{R}^{(1)}$ independently to bricks $x_{1:k}, x_{k+1:2k}, \ldots$ to compute $\nu_k^{(1)}$ for all $k$. The second layer RNN $\mathcal{R}^{(2)}$ produces class labels or in multi-layer case, inputs for the next layer. Note that $\nu_2^{(1)}, \nu_3^{(1)}$ can be reused for evaluating the next window. **(b), (c)**: Mean squared approximation error and the prediction accuracy of ShaRNN with zeroth and first order approximation ($M = 1, 2$ respectively in Claim 3) with different brick-sizes $k$ (for Google-13). Note the large error with $M = 1$ (same as truncation method in [23]). $M = 2$ introduces significant improvement, especially for small $k$, but clearly needs larger $M$ to achieve better accuracy. **(d)**: Comparison of norm of gradient vs Hessian of $\mathcal{R}(h_t, x_{t+1:t+k})$ with varying $k$. $\mathcal{R}$ is FastRNN [21] with swish activation. Smaller Hessian norm indicates that the first-order approximation of $\mathcal{R}$ (Claim 3) by ShaRNN is more accurate than the 0-th order one (ShaRNN with $M = 1$) suggested by [23].

## 3.2   Multi-layer ShaRNN

Above claim shows that selecting small $k$ leads to a large number of bricks and hence, a large number of points to be processed by second layer RNN $\mathcal{R}^{(2)}$ which will be the bottleneck in inference. However, using the same approach, we can replace the second layer with another layer of ShaRNN to bring down the cost. By repeating the same process, we can design a general $L$ layer architecture where each layer is equipped with a RNN model $\mathcal{R}^{(l)}$ and the output of a $l$-th layer brick is given by:

$$\nu_j^{(l)} = \mathcal{R}^{(l)}([\nu_{(j-1)\cdot k+1}^{(l-1)}, \ldots, \nu_{(j-1)k+k}^{(l-1)}]),$$

for all $1 \leqslant j \leqslant T/k^l$, where $\nu_j^{(0)} = x_j$. The predicted label is given by $\hat{y} = f(\nu_{T/k^{L-1}}^{(L)})$.

Using argument similar to the claim in the previous section, we can reduce the total inference cost to $O(\log T)$ by using $k = O(1)$ and $L = \log T$.

**Claim 2.** *Let all layers of multi-layer ShaRNN have same hidden-size and per-time step complexity $C_1$ and let $k = \omega$. Then, the additional cost of applying ShaRNN to $X^{s+1}$ is $O(T/k^L + L \cdot k) \cdot C_1$, where $X^s = x_{(s-1)\cdot\omega+1:(s-1)\cdot\omega+T}$. Consequently, selecting $L = \log(T)$, $k = O(1)$, and assuming $\omega = O(1)$, the total amortized cost is $O(C_1 \cdot \log(T))$.*

That is, we can achieve exponential speed-up over $O(T)$ cost for standard RNN. However, such a model can lead to a large loss in accuracy. Moreover, constants in the cost for large $L$ are so large that a network with smaller $L$ might be more efficient for typical values of $T$.

## 3.3   MI-ShaRNN

Recently, [10] showed that several time-series training datasets are coarse and the sliding window size $T$ can be decreased significantly by using their multi-instance based algorithm (MI-RNN). MI-RNN finds tight windows around the actual signature of the class, which leads to significantly smaller models and reduces inference cost. Our ShaRNN architecture is orthogonal to MI-RNN and can be combined to obtain even higher amount of inference saving. That is, MI-RNN takes the dataset

$\mathcal{Z} = \{(X_1, y_1), \ldots, (X_n, y_n)\}$ with $X_i$ being a sequential data point over $T$ steps and produces a new set of points $X'_j$ with labels $y'_j$, where each $X'_j$ is sequential data point over $T'$ and $T' \leqslant T$. MI-ShaRNN applies ShaRNN to the output of MI-RNN so that the inference cost is dependent only on $T' \leqslant T$, and captures the key signal in each data point.

## 4 Analysis

In this section, we provide theoretical underpinnings of ShaRNN approach and we also put it in context of work by [23] that discusses RNN models for which we can get rid of almost all of the recurrence.

Let $\mathcal{R} : \mathbb{R}^{d+\hat{d}} \to \mathbb{R}^{\hat{d}}$ be a standard RNN model that maps the given hidden state $h_{t-1} \in \mathbb{R}^{\hat{d}}$ and data point $x_t \in \mathbb{R}^d$ into the next hidden state $h_t = \mathcal{R}(h_{t-1}, x_t)$. Overloading notation, $\mathcal{R}(h_0, x_1, \ldots, x_t) = \mathcal{R}(h_{t-1}, x_t)$. We define a function to be recurrent if the following holds:
$$\mathcal{R}(h_0, x_1, \ldots, x_t) = \mathcal{R}(\mathcal{R}(h_0, x_1, \ldots, x_{t-1}), x_t).$$
The final class prediction using feed-forward layer is given by: $\hat{y} = f(h_T) = f(\mathcal{R}(h_0, x_{1:T}))$. Now, ShaRNN attempts to untangle and approximate the dependency of $f(h_T)$ and $\mathcal{R}(h_0, x_{1:T})$ on $h_0$, by using Taylor's theorem. Below claim shows the condition under which the approximation error introduced by ShaRNN is small.

**Claim 3.** *Let $\mathcal{R}(h_0, x_1, \ldots, x_t)$ be an RNN and let $\|\nabla_h^M \mathcal{R}(h, x_{t:t+k})\| \leqslant O(\epsilon \cdot M!)$ for some $\epsilon \geqslant 0$ where $\nabla_h^M$ is $M$th order derivative with respect to $h$. Also let $\|\mathcal{R}(h_0, x_{1:t}) - h_0\| = O(1)$, $\|\nabla_h^m \mathcal{R}(h_0, x_{t+1:t+k})\| = O(m!)$ for all $t \in [T]$. Then, there exists an ShaRNN defined by functions $\mathcal{R}^{(1)}$, $\mathcal{R}^{(2)}$ and brick-size $k$, s.t.:*
$$\|\mathcal{R}^{(2)}(\nu_1^{(1)}, \ldots, \nu_{T/k}^{(1)}) - \mathcal{R}(h_0, x_{1:T})\| \leqslant \epsilon \cdot M \cdot T, \text{ where } \nu_j^{(1)} = \mathcal{R}^{(1)}(h_0, x_{(j-1) \cdot k+1:j \cdot k}).$$

See Appendix A for a detailed proof of the claim.

The above claim shows that the hidden state computed by ShaRNN is close to the state computed by a fully recursive RNN, hence the final output $\hat{y}$ would also be close. We now compare this result to the result of [23], which showed that $\|\mathcal{R}(h_0, x_{1:T}) - \mathcal{R}(h_0, x_{T-k+1:T})\| \leqslant \epsilon$ for large enough $k$ if $\mathcal{R}$ satisfies a contraction property. That is, if $\|\mathcal{R}(h_{t-1}, x_t) - \mathcal{R}(h'_{t-1}, x_t)\| \leqslant \lambda \|h_{t-1} - h'_{t-1}\|$ where $\lambda < 1$. However, $\lambda < 1$ is a strict requirement and do not hold in practice. Due to this, if we only compute $\mathcal{R}(h_0, x_{T-k+1:T})$ as suggested by the above result (for some reasonable values of $k$), then resulting accuracy on several datasets drops significantly (see Figure 1(b),(c)).

In the context of Claim 3, result of [23] is a special case with $M = 1$, i.e., the result only applies a $0-$th order Taylor series expansion. Figure 1 (d) shows how norm of the gradient that bounds error due to the 0-th order expansion is significantly larger than the norm of the Hessian which bounds error due to the 1-st order expansion.

**Case study with FastRNN**: We now instantiate Claim 3 for a simple FastRNN model [21] with a first-order approximation i.e., with $M = 2$ in Claim 3.

**Claim 4.** *Let $\mathcal{R}(h_0, x_1, \ldots, x_t)$ be a FastRNN model with parameters $U, W$. Let $\|U\| \leqslant O(1)$, $\|\nabla_h^2 \mathcal{R}(h_0, x_{t:t+k})\| \leqslant O(\epsilon)$ for any $k$-length sequence. Then, there exists an ShaRNN defined by functions $\mathcal{R}^{(1)}$, $\mathcal{R}^{(2)}$ and brick-size $k$ s.t.: $\|\mathcal{R}^{(2)}(\nu_1^{(1)}, \ldots, \nu_{T/k}^{(1)}) - \mathcal{R}(h_0, x_{1:T})\| \leqslant \epsilon$, where $\nu_j^{(1)} = \mathcal{R}^{(1)}(h_0, x_{(j-1) \cdot k+1:j \cdot k})$.*

Note that $\|U\| = O(1)$ holds for all the benchmarks that were tried in [21]. Moreover, this assumption is significantly weaker than the typical $\|U\| < 1$ assumption required by [23]. Finally, the Hessian term is significantly smaller than the derivative term (Figure 1 (d)), hence the approximation error and prediction error should be significantly smaller than the one we would get by 0-th order approximation (see Figure 1 (b), (c)).

## 5 Empirical Results

We conduct experiments to study: a) performance of MI-ShaRNN with varying hidden state dimensions at both the layers $\mathcal{R}^{(1)}$ and $\mathcal{R}^{(2)}$ to understand how its accuracy stacks up against baseline

Table 1: Table compares maximum accuracy achieved by each of the method for different model sizes, i.e., different hidden-state sizes indicated by numbers in bracket; MI-ShaRNN reports two numbers for the first and the second layer, respectively. Table also reports the corresponding computational cost (amortized number of flops required per data point inference) for each method. $T$ denotes the no. of time-steps for the dataset, $T'$ denotes the trimmed number of time-steps obtained by MI-RNN, $k$ is the selected brick-length for MI-ShaRNN. Note that for all but one datasets MI-ShaRNN is able to achieve similar or better accuracy compared to baseline LSTMs.

| Dataset | Baseline LSTM | | | MI-RNN | | | MI-ShaRNN | | |
|---|---|---|---|---|---|---|---|---|---|
| | Acc(%) | Flops | $T$ | Acc(%) | Flops | $T'$ | Acc(%) | Flops | $k$ |
| Google-13 | 91.13 (64) | 4.89M | 99 | 93.16 (64) | 2.42M | 49 | **94.01** (64, 32) | 0.59M | 8 |
| HAR-6 | 93.04 (32) | 1.36M | 128 | 91.78 (32) | 0.51M | 48 | **94.02** (32, 8) | 0.17M | 16 |
| GesturePod-5 | 97.13 (48) | 8.37M | 400 | 98.43 (48) | 4.19M | 200 | **99.21** (48, 32) | 0.83M | 20 |
| STCI-2 | 99.01 (32) | 2.67M | 162 | 98.43 (32) | 1.33M | 81 | **99.23** (32, 32) | 0.30M | 8 |
| DSA-19 | 85.17 (64) | 7.23M | 129 | **88.11** (64) | 5.05M | 90 | 87.36 (64, 48) | 1.10M | 15 |

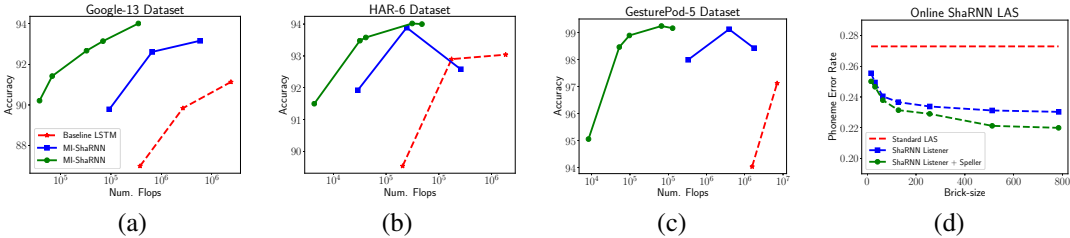

(a)          (b)          (c)          (d)

Figure 2: (a),(b),(c): Accuracy vs inference cost: we vary model size (hidden dimensions) to obtain accuracy vs inference cost curve for different methods. All the three plots show that MI-ShaRNN produces more accurate models with as much as $8-10$x reduction in the inference cost. (d): Error-rate of standard LAS method [12] and of ShaRNN based streaming LAS with varying brick-sizes $k$ on the TIMIT [13] dataset. We report results when both Listener+Speller use ShaRNN vs when only Listener uses it. ShaRNN Listener+Speller with $k = 64$ incurs 12x smaller lag in phoneme prediction vs baseline LAS ($k = 784$).

models across different model sizes, b) inference cost improvement that MI-ShaRNN produces for standard time-series classification problems over baseline models and MI-RNN models, c) if MI-ShaRNN can enable certain time-series classification tasks on devices based on the tiny Cortex M4 with only 100MHz processor and 256KB RAM. Recall that MI-ShaRNN uses ShaRNN on top of trimmed data points given by MI-RNN. MI-RNN is known to have better performance than baseline LSTMs, so naturally MI-ShaRNN has better performance than ShaRNN. Hence, we present results for MI-ShaRNN and compare them to MI-RNN to demonstrate advantage of ShaRNN technique.

**Datasets**: We benchmark our method on standard datasets from different domains like audio keyword detection (Google-13), wake word detection (STCI-2), activity recognition (HAR-6), sports activity recognition (DSA-19), gesture recognition (GesturePod-5). The number after hyphen in dataset name indicates the number of classes in the dataset. See Table 3 in appendix for more details about the datasets. All the datasets are available online (see Table 3) except STCI-2 which is a proprietary wake word detection dataset.

**Baselines**: We compare our algorithm MI-ShaRNN (LSTM) against the baseline LSTM method as well as MI-RNN (LSTM) method. Note that MI-RNN as well as MI-ShaRNN build upon an RNN cell. For simplicity and consistency, we have selected LSTM as the base cell for all the methods, but we can train each of them with other RNN cells like GRU [7] or FastRNN [21]. We implemented all the algorithms on TensorFlow and used Adam for training the models [19]. The inference code for Cortex M4 device was written in C and compiled onto the device. All the presented numbers are averaged over 5 independent runs. The implementation of our algorithm is released as part of the EdgeML [11] library.

**Hyperparameter selection**: The main hyperparameters are: a) hidden state sizes for both the layers of MI-ShaRNN. b) brick-size $k$ for MI-ShaRNN. In addition, the number of time-steps $T$ is associated with each dataset. MI-RNN prunes down $T$ and works with $T' \leqslant T$ time-steps. We provide results

Table 2: Deployment on Cortex M4: accuracy of different methods vs inference time cost (ms) on M4 device with 256KB RAM and 100MHz processor. For low-latency keyword spotting (Google-13), the total inference time budget is 120 ms.

|      | Baseline | | MI-RNN | | MI-ShaRNN | |
|------|-------|-------|-------|-------|----------|---------|
|      | 16    | 32    | 16    | 32    | (16, 16) | (32,16) |
| Acc. | 86.99 | 89.84 | 89.78 | 92.61 | **91.42** | **92.67** |
| Cost | 456   | 999   | 226   | 494   | **70.5**  | **117**   |

with varying hidden state sizes to illustrate trade-offs involved with selecting this hyperparameter (Figure 2). We select $k \approx \sqrt{T}$ with some variation to optimize w.r.t the stride length $\omega$ for each dataset; we also provide an ablation study to illustrate impact of different choices of $k$ on accuracy and the inference cost (Figure 3, Appendix).

**Comparison of accuracies**: Table 1 compares accuracy of MI-ShaRNN against baselines and MI-RNN for different hidden dimensions at $\mathcal{R}^1$ and $\mathcal{R}^2$. In terms of prediction accuracies, MI-ShaRNN performs much better than baselines and is competitive to MI-RNN on all the datasets. For example, with only $k = 8$, MI-ShaRNN is able to achieve 94% accuracy on the Google-13 dataset while MI-RNN model is applied for $T = 49$ steps and baseline LSTM for $T = 99$ steps. That is, with only 8-deep recurrence, MI-ShaRNN is able to compete with accuracies of 49 and 99 deep LSTMs.

For inference cost, we study the amortized cost per data point in the sliding window setting (See Section 3). That is, baseline and MI-RNN for each sliding window recomputes the entire prediction from scratch. But, MI-ShaRNN can re-use computation in the first layer (see Section 3) leading to significant saving in inference cost. We report inference cost as the additional floating point operations (flops) each model would need to execute for every new inference. For simplicity, we treat both addition and multiplication to be of same cost. The number of non-linearity computations are small and are nearly same for all the methods so we ignore them.

Table 1 clearly shows that to achieve best accuracy, MI-ShaRNN is up to 10x faster than baselines and up to 5x faster than MI-RNN, even on a single threaded hardware architecture. Figure 2 shows computation vs accuracy trade-off for three datasets. We observe that for a range of desired accuracy values, MI-ShaRNN is 5-10x faster than the baselines.

Next, we compute accuracy and flops for MI-ShaRNN with different brick sizes $k$ (see Figure 3 of Appendix). As expected, $k \sim \sqrt{T}$ setting requires fewest flops for inference, but the story for accuracy is more complicated. For this dataset, we do not observe any particular trend for accuracy; all the accuracy values are similar, irrespective of $k$.

**Deployment of Google-13 on Cortex M4**: we use ShaRNN to deploy a *real-time* keyword spotting model (Google-13) on a Cortex M4 device. For time series classification (Section 3), we will need to slide windows and infer classes on each window. Due to small working RAM of M4 devices ($256KB$), for real-time recognition, the method needs to finish the following tasks within a budget of 120ms: collect data from the microphone buffer, process them, produce ML based inference and smoothened out predictions for one final output.

Standard LSTM models for this task work on 1s windows, whose featurization generates a $32 \times 99$ feature vector; here $T = 99$. So, even a relatively small LSTM (hidden size 16), takes on 456ms to process one window, exceeding the time budget (Table 2). MI-RNN is faster but still requires 225ms. Recently, a few CNN based methods have also been designed for low-resource keyword spotting [26, 20]. However, with just 40 filters applied to the standard $32 \times 99$ filter-bank features, the working memory requirement balloons up to $\approx 500KB$ which is beyond typical M4 devices' memory budget. Similarly, compute requirement of such architectures also easily exceed the latency budget of 120ms. See Figure 4, in the Appendix for a comparison between CNN models and ShaRNN.

In contrast, our method is able to produce inference in only 70ms, thus is well-within latency budget of M4. Also, MI-ShaRNN holds two arrays in the working RAM: a) input features for 1 brick and b) buffered final states from previous bricks. For the deployed MI-ShaRNN model, with timesteps $T = 49$, brick-size $k = 8$ working RAM requirement is just 1.5 KB.

**ShaRNN for Streaming Listen Attend Spell (LAS)**: LAS is a popular architecture for phoneme classification in given audio stream. It forms non-overlapping time-windows of length 784 ($\approx 8$ seconds) and apply an encoder-decoder architecture to predict a sequence of phonemes. We study

LAS applied to TIMIT dataset [13]. We enhance the standard LAS architecture to exploit time-annotated ground truth available in TIMIT dataset, which improved baseline phoneme error rate from publicly reported $0.271$ to $0.22$. Both Encoder and Decoder layer in standard and enhanced LAS consists of fully recurrent bi-LSTMs. So for each time window (of length $784$) we would need to apply entire encoder-decoder architecture to predict the phoneme sequence, implying a potential lag of $\approx 8$ seconds ($784$ steps) in prediction.

Instead, using ShaRNN we can divide both the encoder and decoder layer in bricks of size $k$. This makes it possible to give phoneme classification for every $k$ steps of points thereby bringing down lag from $784$ steps to $k$ steps. However, due to small brick size $k$, in principle we might lose significant amount of context information. But due to the corrective second layer in ShaRNN (Figure 1) we observe little loss in accuracy. Figure 2 shows performance of two variants of ShaRNN + LAS: a) ShaRNN Listener that uses ShaRNN only in encoding layer, b) ShaRNN Listener + Speller that uses ShaRNN in both the encoding and decoding layer. Figure 2 (d) shows that using ShaRNN in both the encoder and decoder is more beneficial than using it only in encoder layer. Furthermore, decreasing $k$ from 784 to 64 leads to marginal increase in error from $0.22$ to $0.238$ while reducing the lag significantly; from $8$ seconds to $0.6$ seconds. In fact, even at $k = 64$ this model's performance is significantly better than the reported error of standard LAS ($0.27$) [12]. See Appendix C for details.

## Footnotes

*Work done as a Research Fellow at Microsoft Research India.

†Work done during internships at Microsoft Research India.

[3]`https://en.wikipedia.org/wiki/ARM_Cortex-M#Cortex-M4`

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
