[Supplementary Material]

**A Proofs**

*Proof of Claim 1.* Recall that for streaming setting, sliding windows $X^s$ can then be broken intro bricks $B_j = x_{((j-1)\cdot k+1):(j\cdot k)}$ where $(s-1)\cdot q + 1 \leqslant j \leqslant (s-1)\cdot q + T/k$. Now first layer of SRNN compute $\nu_j^{(1)}$ for all $j$. Hence, for the next sliding window $X^{s+1} = x_{s\cdot\omega+1:s\cdot\omega+T}$, we can reuse $\nu_j^{(1)}$ from the previous window where $s\cdot q + 1 \leqslant j \leqslant (s-1)\cdot q + T/k$. Note that the second layer would still need to be computed from scratch. Hence, for new window $X^{s+1}$, we need to compute $\mathcal{R}^{(1)}$ over $\omega = q\cdot k$ new steps. Furthermore, $\mathcal{R}^{(2)}$ needs to be computed over $T/k$ steps. So the total compute requirement is: $\left(\frac{T}{k} + q\cdot k\right)\cdot C_1$. Second part of Claim follows by setting $k = \sqrt{T/q}$. □

*Proof of Claim 3.* Define,

$$\nu_j^{(1)} = vec([\mathcal{R}(h_0, x_{t:t+k-1}); \nabla_h^1 \mathcal{R}(h_0, x_{t:t+k-1}); \ldots; \frac{1}{M!}\nabla_h^{M-1}\mathcal{R}(h_0, x_{t:t+k-1})]), \quad (1)$$

where $t = (j-1)\cdot k + 1$ and $j \in [T/k]$. Using Claim 5, $\nu_j^{(1)}$ is a recurrent function of $h_0$, $x_i$'s, and can be computed by an RNN $\mathcal{R}^{(1)}$ applied to $x_{t:t+k-1}$ and $h_0$.

Similarly, define:

$$\nu_j^{(2)} = \mathcal{R}(h_0, x_{t:t+k-1}) + \sum_{m=1}^{M-1}\frac{1}{m!}\nabla_h^m\mathcal{R}(h_0, x_{t:t+k})\cdot(\nu_{j-1}^{(2)} - h_0)^{\otimes m} +$$

$$\frac{1}{M!}\nabla_h^M\mathcal{R}(\zeta, x_{t:t+k})\cdot(\nu_{j-1}^{(2)} - h_0)^{\otimes M}, \quad (2)$$

where $\nu_0^{(2)} = h_0$. Note that there exists a simple bi-linear function $\mathcal{R}^{(2)}$ s.t. $\nu_j^{(2)} = \mathcal{R}^{(2)}(\nu_{j-1}^{(2)}, \nu_j^{(1)})$.

Using the assumptions mentioned in the Claim, we will now show that $\nu_j^{(2)} \approx h_t$ for SRNN with $\mathcal{R}^{(1)}, \mathcal{R}^{(2)}$ defined above and where $t = j\cdot k$.

Using Taylor's theorem:

$$\mathcal{R}(h_0, x_{1:t+k-1}) = \mathcal{R}(h_0, x_{t:t+k-1}) + \sum_{m=1}^{M-1}\frac{1}{m!}\nabla_h^m\mathcal{R}(h_0, x_{t:t+k})\cdot(h_{t-1} - h_0)^{\otimes m} +$$

$$\frac{1}{M!}\nabla_h^M\mathcal{R}(\zeta, x_{t:t+k})\cdot(h_{t-1} - h_0)^{\otimes M}, \quad (3)$$

where $\zeta = \lambda h_0 + (1-\lambda)h_{t-1}$ for some $\lambda > 0$.

Using triangular inequality:

$$\|\mathcal{R}(h_0, x_{1:t+k-1}) - \nu_j^{(2)}\| \leqslant \|\frac{1}{M!}\nabla_h^M\mathcal{R}(\zeta, x_{t:t+k-1})\| \times \|(h_{t-1} - h_0)^{\otimes M}\| +$$

$$\sum_{m=1}^{M-1}\frac{1}{m!}\|\nabla_h^m\mathcal{R}(h_0, x_{t:t+k-1})\| \times \|(h_{t-1} - h_0)^{\otimes M} - (\nu_{j-1}^{(2)} - h_0)^{\otimes M}\|,$$

where $t = (j-1)\cdot k + 1$. Using the assumptions of claims along with standard algebraic manipulations, we get:

$$\|\mathcal{R}(h_0, x_{1:t+k-1}) - \nu_j^{(2)}\| \leqslant \epsilon + O(M\epsilon)\|\nu_{j-1}^{(2)} - h_{t-1}\|.$$

The claim now follows by applying the above result recursively for all $j \in T/k$. □

**Claim 5.** *If $f$ is a recurrent function, i.e., $f(h_0, x_{t:t+k}) = f(f(x_{t:t+k-1}, h_0), x_{t+k})$. Then, it's higher-order derivatives are also recurrent.*

*Proof of Claim 4.* FastRNN updates hidden state as: $h_t = \alpha\cdot\sigma(Uh_{t-1} + Wx_t + b) + \beta h_{t-1}$ where $\beta \approx 1 - \alpha$, $\alpha = O(1/T)$ and the activation function $\sigma$ is ReLU. Using the updates, we have: $\|h_t - h_{t-1}\| \leqslant \frac{\|U\|+1}{T}\|h_{t-1}\|$, i.e., $\|h_t\| \leqslant \exp(\|U\| + 1)$ for all $t$. Now by assumption $\|U\| = O(1)$, we have: $\|h_t\| = O(1)$ for all $t$. Similarly, $\|\nabla_h\mathcal{R}(h_{t-1}, x_t)\| \leqslant (1 + \frac{\|U\|+1}{T})\|\nabla_h\mathcal{R}(h_{t-2}, x_{t-1})\|$. Using similar arguments as above, we have $\|\nabla_h\mathcal{R}(h_{t-1}, x_t)\| \leqslant O(1)$ for all $t$. Claim now follows by combining Claim 3 with the bounds on $\|h_t\|$, $\|\nabla_h\mathcal{R}(h_{t-1}, x_t)\|$ and $\|\nabla_h^2\mathcal{R}(h_{t-1}, x_t)\|$. □

## B   Additional Empirical Results

| Dataset | #Steps (Baseline) | Feat. Dim. | #Train | #Val | #Test | Source |
|---------|-------------------|------------|--------|------|-------|--------|
| Google-13 | 99 | 32 | 51088 | 6798 | 6835 | URL1 |
| HAR-6 | 128 | 9 | 6220 | 1132 | 2947 | URL2 |
| STCI-2 | 162 | 32 | 42788 | 5223 | 5224 | Proprietary |
| DSA-19 | 129 | 45 | 4560 | 2280 | 2280 | URL3 |
| GesturePod-5 | 400 | 6 | 13432 | 2684 | 2552 | URL4 |
| TIMIT | 784 | 39 | 4389 | 231 | 1680 | URL5 |

Table 3: Dataset details: Source of dataset, the number of timesteps, feature dimension and the number of data points in train, test and validation tests.

| | |
|---|---|
| URL1 | `http://download.tensorflow.org/data/speech_commands_v0.01.tar.gz` |
| URL2 | `https://archive.ics.uci.edu/ml/datasets/human+activity+recognition+using+smartphones` |
| URL3 | `https://archive.ics.uci.edu/ml/datasets/Daily+and+Sports+Activities` |
| URL4 | `https://www.microsoft.com/en-us/research/publication/` |
| | `gesturepod-programmable-gesture-recognition-augmenting-assistive-devices/` |
| URL5 | `https://catalog.ldc.upenn.edu/LDC93S1` |

Figure 3: Accuracy and inference cost vs brick size ($k$) on HAR-6 dataset for a model with hidden dimension 32 at both the layers. Inference cost in terms of number of floating point operations (flops) behaves as expected as show in in (a). The accuracy trend, shown in (b), is tricky at extreme values of $k$. When $k$ is very small, the lower layer is very shallow, while for high values of $k$, the higher layer becomes shallow.

## C   Online LAS with SRNN

Listen-Attend-Spell is a popular end-to-end architecture for transcribing speech with phonemes. The architecture consists of two parts— the listener and the speller. The listener is a pyramidal recurrent network which encodes the filter bank spectra input. The speller uses an attention-based recurrent network to decode the listener output and produces phonemes. Standard LAS transcribes audio input with 784 steps which corresponds to about 8secs worth of audio clip. While standard LAS architecture's phoneme error-rate on the TIMIT dataset is $0.27$ [2], after a few enhancements like dropping a layer and thresholding out predictions with low confidence, we can achieve the baseline error rate of $0.251$.

Now, the LAS architecture is designed to transcribe static input, i.e., where a fixed-length audio clip (of $\leqslant$ 8sec or 784 steps) and does not readily generalize to the streaming setting where the audio data is flowing in continuously. One approach is to form non-overlapping windows of fixed size and

apply LAS on each of them independently. Naturally such a technique would incur a large lag in phoneme predictions. Another approach is to form sliding windows, but in that case it is not clear how to reconcile predictions from the overlapping sliding windows.

We focus on the streaming setting and propose an SRNN based approach for making the LAS architecture streaming, i.e., with predictions with small lag of say $\leqslant 1\text{sec}$ — this has been illustrated in Figure 4. Intuitively, as new batch of audio data arrives, the goal is to process the new batch of data and predict phonemes contained in the batch; note that batch-size should ideally be small so that there is a small lag in prediction. However, as phoneme prediction can be highly contextual, we cannot process every batch independently and would require context from past few batches of audio as well. But, standard LAS architecture is ill-suited for such task, furthermore, naively processing the past few batches would lead to significant computational overhead.

Below we describe our SRNN-based architecture that can appropriately re-use computation to ensure accurate phoneme prediction with a small batch of audio thus ensuring prediction with small lag and low computational cost.

## C.1 Encoder

We replace the bottom two layers of pyramidal encoder by a 3-layer SRNN where the first two layers partitions the input into "bricks" of size $l_f$ while the third layer recaptures the receptive field by processing bottom layer's output via a bi-LSTM. The output of the third layer is the final code/embedding of the input-sequence. Similar to sliding-window streaming setting (Section 3), we can re-use computation from the bricks to process the new $l_f$-sized brick of audio data efficiently.

Figure 4: SRNN based online LAS.

## C.2 Decoder

We replace the attention-based architecture in LAS with an inverted pyramidal decoder — the number of output states for each of the layer in the inverted pyramidal decoder is twice the number of input states. Thus, after two layers of the inverted pyramidal decode we obtain the same number of output states as the input to the encoder. Each of these output states are then processed by an Multilayer Perceptron (MLP) layer to compute the probability distribution over the space of all phonemes.

Applying the above Decoder in the streaming setting will incur significant computation overhead. To alleviate this concern, we again use SRNN to enable re-use of the computation across the decoder layer as was done for the encoder layer. In particular, we use one recurrent network to compute

the 'summary' of all the output states (denoted as $S_i$ in Figure 4) of a given fragment. Another recurrent network processes the past "summaries" $S_{i-1}, S_{i-2}$ and $S_{i-3}$ to produce $H_i$ which is the 'correction' factor for each of the output states ($u_j$ in Figure 4) of the ith fragment. This correction term concatenated with each $u_j$ is the input to a two layer MLP with softmax output over the phonemes.

Hence, the phoneme distribution is obtained for each new input frame/batch and the network is trained using the time aligned phoneme transcription available in the TIMIT dataset. The final prediction by the model is obtained by removing labels predicted with a low confidence (less than a threshold) and collapsing the repeating phonemes.

## C.3   Argument

We first replace the encoder in LAS while retaining the decoder, we see an improvement in the phoneme error rate from 0.251 to 0.240 ($l_f = 64$) by doing this. Using the SRNN encoder, the streaming input can be transcribed every $l_f$ input frames, thus there is no need to wait for the entire speech input. Even though the lag for prediction is reduced, this still involves the attention computation across all the encoder states which is expensive especially when the input speech is long and runs into hours. To avoid this, we replace the decoder with an SRNN decoder where the need for attention is eliminated by predicting a phoneme for each input frame and not just the unique phonemes. With this substitution, we observe a further improvement in the phoneme error rate to 0.238 ($l_f = 64$).

Surprisingly, it turns out that our new architecture is able to better model the phoneme prediction problem. The error rate for the "offline" version of our model, i.e., where $l_f = 784$ is 0.220. This error-rate is significantly better than the rate of 0.251 that we could obtain using enhancements of the standard LAS model.

As noted above, using our SRNN based architecture with $l_f = 64$, we could still achieve error rate of 0.238 which is marginally larger than the best error rate achieved by $l_f = 784$. However, lag in phoneme predictions in $l_f = 64$ case is 12x smaller than the lag incurred by our architecture with $l_f = 784$, i.e., in the offline case.

## Footnotes

[2] [4] did not report results on any publicly available dataset, but this error-rate matches the publicly reported numbers [11]