[Reviews · NeurIPS 2019]

Reviewer 1



Originality: The author propose a novel and general architecture that, to the best of my knowledge, has not been described before. Thus the idea of the "shallow" two layer RNN architecture as well as the accompanying theoretical analysis and experimental results are all novel. Quality: The claims appear correct, although I have some confidence in not having missed important issues only for claim 1 and 2. The experiments are comprehensive and instill confidence in the proposed architecture and theoretical guarantees. The code they provide appears about average for this type of research prototypes. Clarity. Most of the paper is clear and easy to follow. There are however a few typos and sentences that could be improved with some additional proof reading. (See below for some of the typos I spotted) Significance. The simplicity of the method combined with the well-motivated use case of embedded devices with constrained resources mean that I see this paper as a useful contribution, from which many are likely to benefit and thus worthy of NeurIPS. Question and comments: When running over 5 random seeds, what kind of variance is observed? It would be worth mentioning this at least the supp material, to get a sense of the statistical relevance of the results. 46: ensuring a small model size -> I believe the model size would not be smaller than that of a standard RNN, if so the claim appears a bit misleading Claim 1 appears correct as stated, but the formulation is a bit convoluted, in the sense that one typically would be given T and w, and can decide on a k; whereas in the current formulation it appears as if you are given a T and q and can pick an arbitrary k based on that, which is not really the case. Line 199: from this sentence it is not very clear how SRNN is combined with MI-RNN, it would be good to give a little more details given that all results use this model are based on a Shallow extension of MI-RNN. In the same vein the empirical analysis would be a little stronger if the results of SRNN without MI-RNN would be reported too. Minor: 37: standard -> a standard 81: main contributions -> our main contributions 90: receptive(sliding) -> [space is missing] 135: it looks like s starts at 0 where all other indices start at 1; including line 171 where s starts at 0 137: fairly small constant -> a fairly small constant 138: that is -> which is 139: tiny-devices -> tiny devices 152: I would find it slightly more readable if the first index v^{(2)} was 1 instead of T/k; if you need an index at all at this point 154: should be v^{(1)} not v^{(2)} 159: tru RNN -> a true RNN 159: principal -> principle 172: for integer -> for some integer 240: it's -> its 267: ablation study -> an ablation study latency budget of 120ms -> it's not clear to me where this exact limit comes from is it a limit of the device itself somehow? 318: steps of pionts threby 314: ully -> fully In the MI-RNN paper [10] they benchmark against GesturePod-6, where the current paper benchmarks against GesturePod-5, are they different? If so in what way?

Reviewer 2



The authors propose shallow RNNs, an efficient architecture for time series classification. shallow RNNs can be parallelized as the time sequences are broken down into subsequences that can be processed independently from each other by copies of the same RNN. Their outputs are then passed to a similarly structured second layer. Multi-layer SRNN extends this to more than two layers. The paper includes both a runtime analysis (claims 1 and 2) and an analysis of the approximation accuracy of the shallow RNN compared to a traditional RNN. The idea is straight forward but the paper scores very low on clarity. The authors opt for symbol definitions instead of clear descriptions, especially in the claims. The claims are a central contribution of the paper BUT UNNECESSARILY HARD TO PARSE. The implications of the claims are not described by the authors. That's why I scored their significance as low. Here are specific points that are unclear from the paper: l.133-140 Shouldn't the amortized inference cost for each time step be C1 i.e. O(1)? Why would you rerun the RNN on each sliding window? l. 165 The heavy use of notation is distracting from getting an understanding of what window size w and partition size k you usually use. Is usually k larger than w or the other way around? This makes it hard to understand how the SRNN architecture interacts with streaming. When the data is coming in in streams, are the streams partitioned and the partitions distributed or are the streams distributed Claim 1 * You already defined $X^s$. Defining it here again just distracts from the claim. * q is the ration between w and k (hence it depends on k). It is weird that your statement relates k to q which depends on k. please explain. Claim 2 * Choice of k in Claim 2 seems incompatible with Claim 1. In Claim 1 k = O(sqrt(T)) in Claim 2 k = O(1). Claim 3 * What is M? What is $\Nabla^M_h$? Claim 3 and 4 * Are those bounds tight enough to be useful? Given a specific problem, can we compute how much accuracy we expect to lose by using a specific SRNN? * Can we use these bounds together with the runtime analysis of claims 1 and 2 to draw tradeoff between accuracy and inference cost like in Figure 2? To me the strength of this paper is the proposed model and ist implementation on small chips (video in the supplement) as well as the empirical study. I would have been curious for a discussion on how the proposed architecture relates to convolutional networks. It seems to me that by setting w small, k small and L large, you almost have a convolutional network where the filter is a small RNN instead of a typical filter. In the introduction, it is mentioned that CNNs are considered impractical. I am curious; could it be that in the regimes for which the accuracy of SRNN is acceptable (Claims 3 and 4) they are actually also impractical? Complexity similar to CNNs?

Reviewer 3



Overall this is a well-written paper with proper motivation, clear design, and detailed theoretical and empirical analysis. The authors attempt to improve the inference efficiency of RNN models with limited computational resources while keeping the length of its receptive window. This is achieved by using a 2-layer RNN, whose first layer parallelly processes small bricks of the entire time series and the second layer gathers outputs from all bricks. The authors also extend SRNN in the streaming setting with similar inference complexity. One concern about the bound in Claim 1 in the streaming setting: In line 137: w is required to be fairly small constant independent of T. In line 166: w = k * q (w is a multiple of k, and thus k needs to be small constant) In line 173: The bound becomes O(\sqrt{qT} * C_1) iff k=\sqrt{T/q}, which is not o(1). Therefore, I was expecting analysis in practical applications with large T and small w. In SRNN, will the O(T/k) extra memory cost be an issue during inference? The extension of multi-layer SRNN in Section 3.2 provides at least O(log T) inference complexity. The bound here is too ideal, but it would be great to see empirically how SRNN performs by adding more shallow layers. The empirical improvements over LSTM and MI-RNN on multiple tasks are impressing. ==== Thanks for your responses. I have read the rebuttal and other reviewers' comments. I am glad to see about the experimental comparisons to CNNs and the refinement of your claims in the rebuttal, and I think including them in the manuscript or supplementary would better clarify and strengthen this paper. Overall this is a relatively simple yet effective solution to edge computing, which would keep becoming more important.

[Author Response · NeurIPS 2019]

**[7064] Shallow RNN: Accurate Time-series Classification on Resource Constrained Devices**

We thank the reviewers for their comments; we will make the suggested improvements and fix the minor typos.

**Reviewer 1**: *Model size compared to standard RNN (L41)*: We meant that SRNN is able to maintain the same model
size as an RNN while increasing parallelizability.

*GesturePod-5 vs GesturePod-6:* Both refer to the same dataset and this is a typo.

*S-RNN and MI-RNN (L199):* Without MI-RNN also, SRNN performs well compared to baseline (e.g. on Google-13, it
achieves 1% higher accuracy despite 9x reduction in flops). We will add these results and more details in supplementary.

*Latency budget:* This is mostly dependent on the stride length that in turn is task specific. For keyword spotting,
100-150ms strides are known to be sufficient for achieving reasonable accuracy.

*Variance across random seeds:* We observed little variance across seeds. We will add results with confidence interval
bounds in supplementary material.

*T, w and k in Claim 1:* We are indeed given T and $\omega$, and Claim 1 sets $k$ as: $k = T/\omega$ (up to a few minor corner cases).

**Reviewer 2**: *Re-running RNN on each window:* This is standard practise in time-series classification as in this domain
the RNNs usually are trained on fixed length windows and do not handle varying window lengths accurately. For
example, in the keyword spotting problem with the Google-13 dataset, if we run RNN for 1.5 secs instead of 1 sec
windows (length of training windows), the accuracy drops by >5%.

*Streaming SRNN*: For incoming streams, partitions are distributed. We provided intuitive explanation in L159-168. At a
high level, we point out that in the streaming setting SRNN is able to reuse computation (by reusing $\nu_j^{(1)}$'s) causing the
amortized cost to comes down.

*Claim 1*: It claims that for a given $\omega = q \cdot k$, $k = \sqrt{T/q}$ is optimal. That is, set $k = T/\omega$. Hence, if $\omega = \sqrt{T}$ then
$k = \sqrt{T}$ which is the setting when SRNN achieves best speed-up over vanilla sliding window RNNs.

*Claim 2*: Claim 2 is for multi-layer SRNN while Claim 1 is for 2-layer SRNN. $k$ values match for $L = 1$.

*Claim 3*: $\nabla_h^M$ is the $M$-th order derivative wrt $h$; we will define it in the next draft.

*Claim 3, 4*: These claims provide an indication as to why SRNN is able to achieve comparable performance to standard
RNNs in practice despite smaller recurrence. That is, we show that if the $M$-th order derivative of the RNN is small
then a specific version of SRNN can reasonably approximate the fully recurrent RNN. These claims are in contrast
with claims of [22], that require the 1-st order derivative itself to be small. In Figure 1(b), (c), (d) we provide limited
empirical evidence to support our assumption, i.e., we show that 1-st order derivative can be much larger than the 2-nd
order and hence approximation error by non-retrained version of SRNN is also relatively small.

*S-RNN vs CNN*: As mentioned on L296-302, working RAM and computation require-
ment of CNN based solutions, designed specifically for low-powered devices [25,19],
is still too large to fit on devices like the Cortex M4. Here we present a table with more
explicit numbers. Note that *none* of the CNN models satisfy compute requirement of
$\leq 0.15M$ flops on M4 device. The best CNN model that at least satisfies the RAM
requirement ($< 256KB$) is 3% less accurate than SRNN.

| No. Filters | No. Pools | Acc. | Size(KB) | FlOps |
|---|---|---|---|---|
| 10 | 2 | 0.81 | 375.1 | 1.1M |
| 10 | 4 | 0.85 | 90.4 | 1.5M |
| 20 | 2 | 0.83 | 753.3 | 3.9M |
| 20 | 4 | 0.88 | 190.1 | 5.6M |
| 30 | 4 | 0.90 | 299.1 | 12.1M |
| **SRNN** | - | **0.91** | **26.5** | **0.09M** |

*SRNN with small $k$, $\omega$ = CNN?* Intuitively, an RNN even with a small $k$ is more powerful
than CNN as it applies non-linearity $k$ times while a CNN layer applies non-linearity
only once per $k$-sized filter. Furthermore, in practice we observe that $\omega \approx \sqrt{T}$ and
$k \approx \sqrt{T}$, which is larger than typical CNN filters of size $3 - 5$.

**Reviewer 3:** *Bound in Claim 1:* Yes, from the Claim point of view L137 is problematic as in practice generally
$\omega \approx \sqrt{T}$. For $\omega = O(1)$ also, we can get $\sqrt{T}$ amortized cost but that requires a slightly more complicated version of
SRNN which we didn't discuss in this paper for ease of exposition. We can add details of the same in the supplementary.

$O(T/k)$ *extra memory:* Yes extra memory is needed, but with RNNs in the streaming setting, we are latency bound
and memory is a smaller issue. For instance, the MXChip used in the included video has a 256KB RAM where as the
SRNN model's memory requirement is only 26.5KB and the $T/k$ extra memory turns out to be about $6KB$. We can
easily fit the model computation in RAM.

*Multilayer S-RNN:* It is more beneficial than 2-layer SRNN only for very large value of $T$; in the type of problems we
studied $T$ was large but not large enough to require multi-layer SRNN (L188-190).

[Meta-Review · NeurIPS 2019]

The paper proposes a simple and novel shallow two stage RNN (SRNN) architecture which enjoys computational advantages over other RNNs-type models. The authors provide a theoretical justification under weak assumptions that are verified on real-world benchmarks. The reviewers think that the contribution is interesting and could have high impact. The also think that paper is well written and motivated, contains an interesting theoretical analysis and satisfactory empirical analysis. Concerns around clarity and experimentation were successfully addressed during the rebuttal.